# Design of Innovative Clothing for Pressure Injury Prevention: End-User Evaluation in a Mixed-Methods Study

**DOI:** 10.3390/ijerph20186773

**Published:** 2023-09-17

**Authors:** Anabela Salgueiro-Oliveira, Anderson da Silva Rêgo, Paulo Santos-Costa, Rafael A. Bernardes, Luísa Filipe, Liliana B. Sousa, Rochelne Barboza, Miguel Carvalho, Maria Bouçanova, Maria Clara Ferreira da Graça Lopes, João A. Apóstolo, Pedro Parreira

**Affiliations:** 1Health Sciences Research Unit: Nursing (UICISA: E), Nursing School of Coimbra (ESEnfC), 3000-232 Coimbra, Portugalrafaelalvesbernardes@esenfc.pt (R.A.B.); baptliliana@esenfc.pt (L.B.S.); apostolo@esenfc.pt (J.A.A.);; 2Centre for Textile Science and Technology (2C2T), University of Minho, 4800-058 Guimaraes, Portugalmigcar@det.uminho.pt (M.C.); 3Impetus Portugal-têxteis Sa (IMPETUS), 4740-696 Barcelos, Portugal; 4Unidade de Cuidados na Comunidade Norton de Matos (UCCNM), 3030-790 Coimbra, Portugal

**Keywords:** biomedical technology, wounds, ergonomic design, protective clothing, pressure injury, mobility limitation, bedridden persons, 4NoPressure

## Abstract

The global relevance of pressure injury (PI) prevention technologies arise from their impact on the quality of life of people with limited mobility and the costs associated with treating these preventable injuries. The purpose of this mixed methods study is to evaluate the design of a prototype integrating Smart Health Textiles for PI prevention based on feedback from specialist nurses who care for individuals who are prone to or have PIs. This is a mixed methods study. A structured questionnaire was conducted as part of an evaluation of a prototype garment for the prevention of PIs. This questionnaire was applied during the evaluation of the prototype and afterwards focus group discussions were held with experts. Descriptive statistics techniques were used to analyze the data and thematic and integrated content analysis was conducted through concomitant triangulation. Nineteen nurses took part, aged 30 to 39 years (52.6%) and with 12.31 ± 8.96 years of experience. Participants showed that the prototype required more manipulation and physical effort, which interfered its usefulness, in addition to presenting difficulties with the openings and the material of the closure system, which interfered with the ease of use and learning. Overall satisfaction with the product was moderate, with some areas for improvement found, such as satisfaction, recommendations to colleagues, and pleasantness of use. It is concluded that areas for improvement have been found in all dimensions, including in the design of openings and the choice of materials. These findings supply significant insights for improving clothing to meet the needs of healthcare professionals and patients.

## 1. Introduction

Technological advancements have contributed significantly to the research and development of health products, particularly in the preventive context of various morbidities, including pressure injuries (PIs) [1]. PIs, also known as pressure ulcers, are wounds that occur because of prolonged pressure on specific areas of the human body, in combination with other individual clinical factors such as physiological and nutritional status, leading to tissue damage [1,2,3].

This morbidity poses significant challenges to public health, affecting the quality of life for individuals and increasing the workload of healthcare professionals and formal/informal caregivers who strive to prevent their occurrence [4]. In response to this public health issue, innovative advancements in medical devices and biomaterials have been complemented by technological progress. Notably, assistive technologies have been developed as valuable tools to prevent PIs among individuals who are bedridden and/or have limited mobility (BRLM) [5]. Existing technologies include high-density foam mattresses, elastic fibers, and motorized beds, which aid in alternating decubitus to reduce the pressure exerted on specific anatomic areas at elevated risk of developing PIs [6]. These devices also help to reduce humidity and friction, allowing for a better quality of life for this population [7].

A review study by Sikka and Garg [8] describes the critical role of adapted clothing in improving the quality of life of BRLM patients as undeniable. The study finds that the use of clothing especially designed to meet the specific needs of these people can supply significant benefits by reducing the occurrence and severity of PIs. The authors of another study [9] conducted experiments on prototypes with user-centered design and concluded their potential as effective tools for managing and organizing the design process.

In addition, the importance of customized clothing also extends to controlling the skin’s microclimate, as pointed out by Arfah et al. [10]. By adjusting the fabric, design, and characteristics of clothing, it is possible to create a suitable environment for the skin, minimizing excessive perspiration and moisture, factors that can trigger or aggravate PIs [1].

Functional textiles allow the incorporation of biosensors and other electronic devices, which elevates their preventive potential. However, research shows that end users do not use wearable devices due to unfamiliarity with their premise and design interface, and a lack of short-term results [11,12,13]. Thus, usability evaluations from laboratory prototypes using user-centered design can be promising in device development and potential usefulness, ease of use and learning, and user satisfaction [9,14,15,16].

Studies aimed at modeling and designing clothing for people with reduced mobility or who are bedridden need to condition issues related to ergonomic comfort. The definitions of ergonomic and design parameters for this type of clothing imply knowing their physical, social, and psychological characteristics, their limitations, safety, mobility, and interaction with support surfaces, and their hygiene and health care needs [9,16,17,18]. 

The 4NoPressure project [19] aims to investigate and develop a clothing typology of Smart Health Textiles for preventing PIs in people with reduced mobility or who are bedridden, with pressure, temperature, and humidity biosensors [16,19]. The project aims to develop the textile typology in eight phases, involving the application of clinical research and industrial and experimental processes [19]. 

Receiving feedback from end users plays a crucial role in the iteration process, ensuring a design is tailored to their preferences and needs. This approach, also known as “user-centered design”, considers human factors such as comfort, ergonomics, aesthetics, and overall usability, resulting in a functional, attractive, and pleasant device [9,17,20,21,22,23,24]. The purpose of this mixed methods study is to evaluate the design of a prototype integrating Smart Health Textiles for PI prevention based on feedback from specialist nurses who care for individuals who are prone to or have PIs.

## 2. Materials and Methods

### 2.1. Study Design

This study adopts a mixed research approach, employing the strategy of concomitant data triangulation based on the constructivist approach and following the guidelines of the Mixed Method Appraisal Tool (MMAT) guide [25]. A mixed research method is justified by comprehensively understanding the phenomenon under investigation and finding critical convergences, divergences, and combinations between the quantitative evaluation and the qualitative insights derived from the focus group (FG) technique. 

The User-Centered Design approach was employed in this study, which involves actively involving the user in the design process [26,27]. This approach aims to understand the needs and characteristics of the users to improve the product being developed. As part of this approach, an exploratory study was conducted to assess the need for potential pajamas to reduce PIs in persons with BRLM [16,19]. 

Based on these requirements, a prototype of the pajamas was created and then evaluated through FG sessions. These sessions were held to evaluate the interaction between professionals and the effectiveness of different prototyping methods. Throughout the development process, each prototype was evaluated by the participants, following the recommendations of an interactive and continuous design process, considering the attributes of relative advantage, compatibility, complexity, and experimentation [20,21,22,23,24,26,27].

### 2.2. Study Setting

The research was conducted within the scope of the Health Sciences Research Unit: Nursing (UICISA: E), and data collection took place at the facilities of the Nursing School of Coimbra (ESEnfC) in Portugal. The recruitment of participants and the step-by-step implementation of the methodological approach (QUAN + QUAL) were started according to the planned strategy and protocol of the 4NoPressure project [19].

### 2.3. Participants and Recruitment

The recruited professionals had extensive knowledge, skills, and experience in healthcare for people with BRLM. To be considered specialists, the professionals had to meet at least two of the following criteria: to have developed specific prevention and health promotion actions for people with BRLM, to have a post-graduate qualification in the area, or to have clinical experience or scientific papers related to the theme. 

A list was compiled with all the possible experts for participation, and an active search was conducted to obtain the contacts, followed by sending invitations through letters. Nineteen nurses with experience in teaching, research, and clinical practice in rehabilitation, intensive care, community and home care services, long-term care, and residential and nursing facilities for the elderly took part in the evaluation process, according to their responses to the invitation sent.

### 2.4. Description of the Prototype 

The modeling and design of pajamas for people with reduced mobility are essential features to meet the specific needs of this group and supply a more adequate and functional clothing experience. These features were created based on a series of reasons and justifications that aimed to maximize comfort, accessibility, and autonomy for the users [16,18,28,29].

The project proposal incorporated several relevant requirements, including design concepts, the structural system, and the overall conception, aimed at achieving ergonomic comfort. These requirements encompassed aspects such as ease of dressing, undressing, and access for personal hygiene. The project also sought to identify the level of effort required for manipulation, the necessary movements during use, and the comfort values to optimize user experience. Moreover, the proposal aimed to accommodate essential items while ensuring easier access to specific body regions [18].

This article primarily focuses on Technology Readiness Levels (TRLs) 1, 2, and 3. These levels involve exploring basic principles and conducting conceptual and analytical testing to align the technology under development with its intended final output [30,31,32,33,34].

For the initial phase, the anthropometric characterization study was carried out, with body measurements made via direct and indirect manual means using 3D Body Scanner technology, with prior delimitation of measurement variables (length and perimeter) and definition of the main resting positions: dorsal decubitus (Position 1), lateral decubitus (Position 2), and sitting position (Position 3) (Figure 1) [18,35,36].

Body measurements were taken to find the proper spacing to maximize patient ergonomic comfort. Defining the best spacing is crucial in ensuring patients’ comfort. By carefully considering anthropometric measurements, it becomes possible to find the ideal amount of space between the body and the fabric of the pajamas. This helps prevent excessive pressure on the contact surface and discomfort for the wearer, while allowing for greater freedom of movement, which is particularly important for individuals with reduced mobility [35,36,37,38].

Based on the anthropometric parameters collected, a two-piece pajama set was developed. The jersey features easy access to the relevant areas and allows the manipulation of medical devices on the arms, eliminating the need to remove the entire pajama set. In addition, an opening was added at the bottom to help with access and personal hygiene care [18,35,36,37,38,39,40]. 

In the case of the pants, openings were incorporated for access to specific regions where medical devices such as bladder probes are found and for a better fit to the body and in consideration of the relative issues of putting on and taking off the pajamas. This decision was made to avoid the accumulation of fabric, which could lead to excessive pressure and discomfort, contributing to the development of pressure injuries [35,36,41,42]. 

The pattern design process used in this study integrated three complementary methodologies: two-dimensional and three-dimensional, using 3D Computer-Aided Design (CAD) software from Modaris 3D Fit by Lectra, Paris, France. The strategic placement of openings is another essential feature of pajama design, supplying greater comfort during rest [35,36,42,43,44].

These openings eliminate the need to remove the pajamas, enabling medical interventions and procedures without causing added patient discomfort. Moreover, these openings promote independence and autonomy for individuals with reduced mobility, allowing them to conduct daily tasks without constant aid [35,36,38,42,43,44].

The 3D CAD system CLO Virtual Fashion (CLO) 3D was used to simulate an avatar with the shape and measurements of a participant based on 3D captures obtained using the Structure Sensor 3D body scanner. It can be inferred that the simulation using the CLO 3D software supplied a better understanding of the pattern design performance, volumetry, drape, shape, and fit [45,46]. 

The software supplies a Tension Map (Strain Map) that gives the degree of fabric stretching during use, which is the tension experienced by the fabric in various positions. Blue shows 100% (no distortion), while red shows a 120% distortion rate. Intermediate values are represented by color gradients (Figure 2) [47,48,49]. 

As an alternative to closing the openings, we used mesh tape composed of recycled material [50]. The choice of suitable materials proves our concern for the aesthetic appearance, environmental impact, and end-user comfort, ensuring a soft and comfortable texture against the users’ skin [42,50,51,52,53].

The proposed design of pajamas for the prevention of PIs incorporates functional and innovative features to improve patient comfort and medical procedures. The pajama set consists of a top and bottom, both designed with strategic openings. The top features front openings to expose the chest and abdomen, allowing easy access to medical areas and avoiding the need to remove the entire pajama (Figure 3).

In addition, openings along the sleeves allow for the hassle-free manipulation of medical devices on the arm. The trousers are equipped with leg and hip openings for similar purposes, and traditional seams are repositioned to reduce friction and prevent pressure ulcers during lateral decubitus. The use of a ribbed knit structure ensures flexibility and a better fit to the body (Figure 3).

### 2.5. Data Collection Instruments and Procedures

Data collection occurred at the UICISA: E during two FG held on 27 and 28 June 2022. The participants were provided with a contextualization of the aim of the FG and were asked to sign the informed consent form for participation in and recording of the discussions.

For the experimentation and evaluation phase of the prototypes developed, a phantom was used. This allowed the evaluators to observe the practical implementations and advantages of the prototypes compared to traditional pajamas, considering the principles of user-centered design. The participants based their evaluation on interactions with the prototypes, including the ability to change lying position, as shown in Figure 1 and Figure 2, and to perform actions such as putting on and taking off the product [20,21,22,23,24,26]. 

This evaluation considered the suitability of the prototypes for typical needs, their complexity, and their relative usefulness in a clinical context, all within the framework of user-centered design. Since this experimentation took place using a phantom model, the simulations carried out provided information on the adjustments or adaptations required [20,21,22,23,24,26,27]. For the quantitative evaluation process, a medical device usability questionnaire was adapted and divided into four sections: utility, consisting of 12 questions about the practicality of using the pajamas; ease of use, consisting of 10 questions; ease of learning, with six questions; and satisfaction/intention to use, consisting of 14 questions related to the participant’s perception of the presented prototype. The questionnaire used a Likert scale ranging from 1—“I strongly disagree” to 7— “I strongly agree”. The responses were collected from health professionals working in various care areas [16,54,55,56,57,58].

After completing the questionnaire for the prototype evaluation, the FG technique [59] was employed to collect qualitative data. During the FG sessions, participants engaged in discussions and reflections on the utility, ease of dressing/undressing, comfort, and physical characteristics of the pajamas, and the implications of their use in caring for the target population. The FG script was designed to include open-ended questions that were randomly organized, considering the ongoing debate and evolving opinions. The questions were adapted to evaluate the prototype based on the established aims [59]. The FG script allowed participants to express their own experiences with a focus on the thematic context, such as in the following:-*How likely are you to adopt the prototype of pajamas for PI prevention in home environments, residential facilities for the elderly, long-term care units, or hospitals?*-*About the characteristics of the presented prototype pajamas, what aspects should be improved and why?*-*What should be the number of openings and the distance between the pieces composing the pajamas? What are the reasons behind these choices?*-*What are the advantages and disadvantages of each of the presented prototypes?*-*How do you perceive the safety of the presented pajama prototypes for individuals with reduced mobility?*

The interviews were conducted by a panel of three moderators, who used a topic-centered approach, emphasizing specific experiences and situations, especially those that expressed prototype evaluation. The moderators enabled the exploration of the findings that appeared through the group discussions and observations, resulting in new insights that led to new questions about the product being evaluated [60].

### 2.6. Data Analysis and Treatment Procedure

The MAXQDA^®^ mixed data analysis software, version 2020, was used to ease the organization of the data analysis phases related to the methodological approach. The transcribed interviews were analyzed using the content analysis technique, thematic modality, which comprised three moments: pre-analysis, or the organization of the existing material for analysis in a flexible way; exploration of the material, covering the identification of communication, which is the moment in which categorization emerges; and the treatment of results, inference, and interpretation, which results in inferences and interpretations of all statements [61]. 

Quantitative data were analyzed using SPSS software, version 24, presenting median, considering that the data value was sorted in ascending order, and confidence intervals to understand variability. Demographic data were presented with mean and standard deviations. QUAN + QUAL data methods were combined through simultaneous triangulation to find points of convergence, divergence, and possible combinations [62].

## 3. Results

This study involved 19 nurses aged 35.36 ± 9.68 years. An almost equal gender distribution (52.6% female, 47.4% male) took part in this study, and 36.8% of the participants had a master’s degree. The mean length of experience in the profession was 12.31 ± 8.96 years, and the mean time in the current unit was 7.15 ± 8.61 years (Table 1).

The nurses’ assessment of the usefulness of the prototype scored a median of 3.0 points. The results showed variations in levels of feeling among research participants. The questions “Allows me to complete the task(s) quickly” and “Allows me to respond to my needs” had medians of 3.0 points, which were the lowest. Participants clearly expressed dissatisfaction with the quantity and hardness of the fastenings used in the pajamas. This signals frustration with managing the garment and suggests that the press studs may be an obstacle to effective and efficient use. These results point to the need to re-evaluate the fastening system in the future development of the pajama prototype (Table 2). 

The convergence of the results suggests that the utility of the garment could be significantly improved by reevaluating and possibly modifying or eliminating the push buttons system. The integrations/inferences present clear information for future iterations of the garment’s design to improve its usability and, so, its effectiveness in preventing PIs.

The quantitative results in the “Ease of use” dimension obtained a median of 3.0 points. The following questions obtained lower medians: “It requires little manipulation to achieve what is intended” (3.0), “It requires few manipulations to achieve what is intended” (3.0) and “It is simple to use” (3.0). The participants discussed and evaluated how the prototype required more manipulations to dress the person with BRLM and required moderate physical effort during use. In addition, they considered the pajama closure system to be not as simple to use as they would like. (Table 3).

The data converge when considering the median of the “Ease of use” dimension and the individual questions within this dimension, which point to a moderate evaluation and highlight the need for improvements in the prototype. The need for enhancement and improvements in the closing system, specifically in the fastenings used, can directly influence the other dimensions when considering the feelings of the evaluators.

Regarding the “Ease of learning” dimension, the median score was 5.0 points, which suggests that participants found the prototype reasonably easy to learn to use. However, the questions “You quickly learn how to use it”, “You easily learn how to use it” and “I quickly became skillful using it” obtained median scores with low points, indicating that participants faced difficulties in quickly learning how to use the garment and becoming skillful with it (Table 4).

The qualitative comments highlighted some concerns about the practicality of the prototype, which may interfere with the ease of learning how to use the pajamas. Participants felt that the garment did not make it easy to change and that some parts of the design were confusing or unnecessary. 

These results converge on the QUANT + QUAL approaches and suggest that while the overall ease of learning is reasonably high, there are specific areas where the garment could be improved to make it easier to use and learn. Specifically, the rapid learning process and the ability to use the garment could be areas of focus for future improvements. In addition, the comments on practicality highlight that aspects of the physical design of the garment could be changed to make it easier to use.

The results presented in Table 3 show that, overall, the participants were moderately satisfied with the product, with an evaluation of 4.0 points in the dimension “Satisfaction/intention to use”. However, there are still areas that need improvement, as the questions “Would be satisfied”, “Would recommend to colleagues”, and “Would be pleasant to use” recorded lower averages. The analysis of the interviews highlighted safety concerns regarding the product, particularly related to the openings and hardness of the push buttons used in the fastening system. Participants suggested that the buttons should be more pliable and spaced out to avoid increasing pressure in injury zones (Table 5).

They also noted that the current closure system makes the garment challenging to wear and expressed dissatisfaction with the presence of a cuff on the pants. Additionally, there were concerns about the size of the clothing, with some individuals finding the pieces too small. Participants also criticized the current design of the openings and provided specific suggestions for improvement, aiming for a more suitable product for their clinical practice.

The QUANT + QUAL results say that while the prototype is potentially helpful and has positive aspects, significant areas must be addressed to improve user satisfaction and intention to use. These include revising the design of the push buttons and openings and possibly adjusting the clothing size. The combination of these results from the two approaches also raised the question about the device’s safety, which may be affected by the closure system and push buttons adopted in the evaluated prototype.

## 4. Discussion

The usability evaluation results of the prototype highlighted that the “usefulness” dimension had an overall average below the recommended level, suggesting the need for changes. These findings were influenced by the questions “It allows me to complete tasks quickly” and “It allows me to satisfy my needs”. These results align with earlier studies that focused on developing clothing methods for elderly individuals with restricted mobility, emphasizing the importance of usability [63,64].

Earlier research has specifically examined a clothing assessment method for elderly individuals with limited mobility, emphasizing the importance of convenient dressing and undressing. Earlier authors have highlighted the significance of individuals’ independence in autonomously using clothing despite physical limitations in mobility [64]. Autonomy issues are attributed to the effort and time needed to put on a garment based on individual needs and proper sizing [64,65,66,67].

In the case of bedridden individuals, clothing utility issues become more complex. Dressing and undressing a person in bed is a daily task for nursing professionals and caregivers of dependent individuals. Hygiene and preventive care activities for pressure injuries involve the movement and manipulation of the wearer’s clothing in bed [67,68]. Improperly designed clothing in terms of wearability can increase the workload for these healthcare professionals and caregivers [69], which may affect the quality of care and patient safety [70].

In the hospital setting, particularly in PI prevention, an earlier study revealed that nurses face challenges managing the care of BRLM patients. Although they recognize the importance of supplying quality care, they often face limitations due to recurring priorities in the clinical setting. The study emphasized the needs of individuals with reduced mobility, highlighting that meeting their complex care needs requires adequate time, human resources, and equipment [68,70].

The lower mean score obtained for the ease of dressing the patient in the assessed prototype pajamas suggests that they may not meet the practical needs of the nursing team, affecting their usefulness. Therefore, the feedback from prototype evaluators should be considered during the redesign stage, followed by a new evaluation of its utility in a clinical setting, focusing on the effectiveness of the developed wearable device [67].

The “Ease of use” dimension of the prototype received a moderate evaluation. Participants highlighted the need for more manipulations to dress BRLM individuals, and the moderate physical effort needed during use. They also found the closure system of the pajamas to be complex. These issues relate to efficiency and effectiveness, which can limit the practicality of the clothing [67,71].

A study conducted in Portugal with individuals diagnosed with Parkinson’s disease, who often have reduced or impaired mobility and are at risk of developing pressure injuries [68], revealed that caregivers and rehabilitation professionals do not use assistive devices such as wearables. Respondents expressed unfamiliarity with the interface design of such devices, which could affect their ease of use and reduce their adoption [72].

Preliminary results from a study in Brazil emphasized that participants perceive comfortable clothing as having better wearability, mainly regarding ease of use. The authors discussed the importance of the ease of learning to wear clothing as a significant aspect of fashion consumption. However, they noted a lack of investment in innovative approaches to clothing for elderly individuals and those with reduced mobility in the current market, highlighting the need for further exploration in this area [73].

In the “Ease of Learning” dimension, the overall average was considered satisfactory but needed improvement. Participants found the prototype reasonably easy to learn to use but noted that it took longer to dress and undress individuals. This limitation hinders the ease of use of the garment and increases the dissatisfaction and reluctance to use the developed medical device. These findings align with Marteli’s study [74], which found design inadequacies as barriers to adopting assistive devices, requiring added time and effort. 

Another study in Spain evaluated a sensorized wearable device that uses a high-frequency Lilypad microcontroller to check vital signs and fluid excretion. The study had already conducted a first approach to evaluate the prototype, in which it was requested to change the raw materials of the drivers used since they were ergonomically unsuitable. As Luna-Perejón et al.’s study [67] highlighted, the second usability evaluation emphasized the main challenges in using the created device, related explicitly to hygiene care practices and manually inputting data into the application. 

Our findings align with Luna-Perejón et al.’s research [67] and further emphasize the difficulties in manipulating the device’s openings and closures. In their study, the evaluated prototype revealed that the materials used in the pajama closure system, such as plastic buttons, proved to have reduced functionality when pressed, compromising the task’s accuracy [67].

A study conducted in Brazil explored the feelings of elderly people with reduced mobility. The understanding presented by the authors refers to the aesthetic, functional, and structural characteristics of the fastening systems. The participants evaluated nine trims, and the use of push buttons as a fastening system stood out. The authors reinforced that the intention to use buttons was not related to adaptive functionality issues but to familiarity with the material due to the historical and cultural context of use. Nevertheless, it was mentioned that, even with the good acceptance of buttons, they introduce stiffness and roughness [75].

It is important to note that individuals with reduced mobility and who are bedridden require more frequent clothing changes and hygiene routines [74]. Clothing that involves the complex manipulation of accessories in its design hinders its practicality and reduces its effectiveness and efficiency in terms of utility. This interaction complexity between patients, caregivers, and healthcare professionals affects the overall performance [67,74,75].

The choice of the best garment fastening system depends on the context of use and the physical and cognitive conditions of the individuals involved. In a hospital setting, fastening systems that are easy to use are necessary due to the high number of patients. Increased manipulation requirements, such as multiple button closures, result in longer dressing times and increased professional workload. The time needed for push closures or complicated fastening systems reduces the ease of use and learning, leading to dissatisfaction and a lower likelihood of intention to use [69,73].

The dimension of satisfaction/intention to use received negative evaluations, particularly regarding product satisfaction. This negative feeling can discourage users from recommending it to others due to its lack of pleasantness and limited usefulness. The evaluation in this dimension may have been influenced by concerns raised in the dimensions of usefulness, ease of use, and ease of learning, which affect interest in using the device due to low satisfaction levels. 

These findings align with Youn and Lee’s study [76], which aimed to propose a model of wearable technology acceptance. The study concluded that difficulties in ease of use could lead to reduced purchase intention or satisfaction with the product. Comparable results were presented in Ju and Lee’s study [77], where respondents reported resistance to buying bright clothes due to low knowledge about their functionality and the perceived complexity of use.

A study [78] evaluated clothing in two groups with mobility difficulties. Participants in the study mentioned fabric type and design as essential factors, favoring those that offered better functionality for dressing and needed less manipulation. The data from Marteli’s study [78] supports the validation of the prototype in this study, as a decrease in ease of use was perceived due to the openings, which can directly affect satisfaction and interest in use.

Studies emphasize the importance of end-user acceptance in terms of satisfaction and intention to use. The concept of intelligent clothing needs to be communicated and tailored to the target population, considering performance and effort expectations [79], easing conditions [80], perceived trust [81], device effectiveness, and usability, all of which positively influence intention to use [82].

The use of 3D body scanning technology to obtain the anthropometric profile of the target group was a crucial aspect of the pajamas’ design. This approach allows precise, customized measurements, considering each person’s body shape and proportions. As a result, pajamas can be custom designed, ensuring a perfect fit and avoiding discomfort caused by clothes that are too baggy or tight [18,35,47,48,49,83].

A study conducted in China [84] employed similar digital technologies, tailoring their product to meet the consumer’s personalized needs. Additionally, they developed computational tools for perfecting the 3D reverse design method using machine learning. The authors concluded that the method created could significantly advance manufacturing, contributing to economic and environmental sustainability and ensuring precision in clothing quality [84]. These advancements can be used in deciding the final prototype, considering the evaluations of professionals who took part in this study.

The study has limitations as it was only evaluated by health professionals, without the involvement of carers and patients. This stage will be carried out in pre-clinical trials after the reformulations have been presented and new materials have been incorporated that meet the needs of the target population. However, the simulated evaluation of the use of the prototypes in phantom, carried out in a laboratory environment, resulted in indicators that can guide Biomedical Textile Engineering researchers towards the potential for involving end users in the development of a new medical device that fulfills their expectations. Another significant limitation is related to the images of the evaluated product. The study project [19] has filed a patent application and is awaiting the opinion of the regulatory agencies.

## 5. Conclusions

The evaluated prototype requires design modifications to enhance the overall usability feeling among nurses. Moderate scores were seen for ease of use and learning, emphasizing the need to improve the openings and materials used in the closure system to meet performance expectations. These aspects relate to the effort needed for product usage and the associated learning process. Nurses reported moderate satisfaction with the product in the “Satisfaction/intention to use” dimension. This moderate level of satisfaction may influence their willingness to integrate the device into their care delivery practices and recommend it to other healthcare professionals, as well as to patients and their families. It is important to note that while participants expressed moderate satisfaction with the product, they also expressed concerns about its safety and practicality, highlighting the need for further pre-clinical studies before going ahead to clinical trials.

## 6. Patents

Modality Provisional Patent Application: PIJAMA FOR THE PREVENTION OF PRESSURE INJURIES, registered on 28 June 2023 in co-ownership with IMPETUS PORTUGAL—TÊXTEIS SA, UNIVERSIDADE DO MINHO, ESCOLA SUPERIOR DE ENFERMAGEM DE COIMBRA, (registration code 118789).

## Figures and Tables

**Figure 1 ijerph-20-06773-f001:**
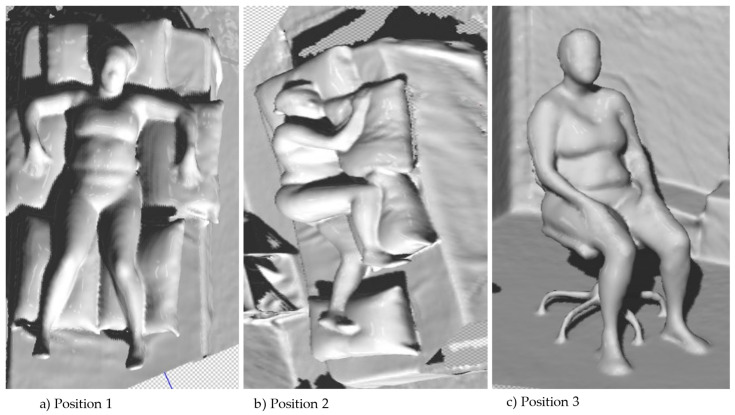
Resting positions used to obtain body measurements and shape: dorsal decubitus (Position 1), lateral decubitus (Position 2), and sitting position (Position 3). Coimbra, Portugal, 2023.

**Figure 2 ijerph-20-06773-f002:**
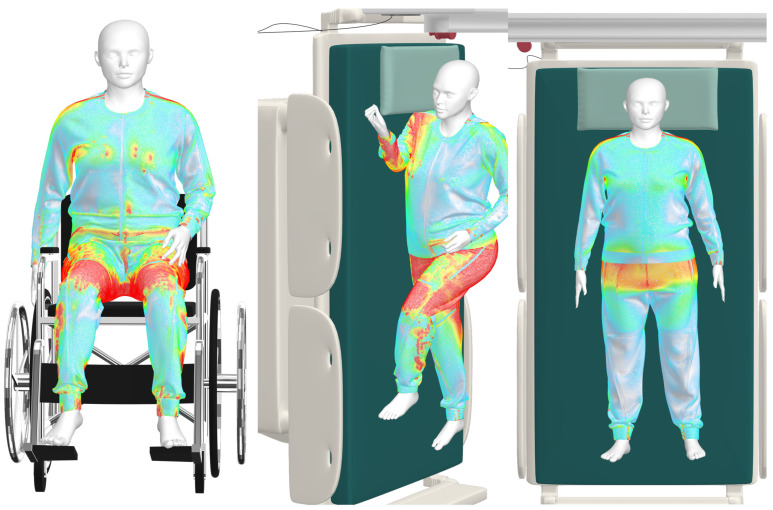
CLO 3D Strain Maps for supine, lateral decubitus, and sitting positions. Coimbra, Portugal, 2023.

**Figure 3 ijerph-20-06773-f003:**
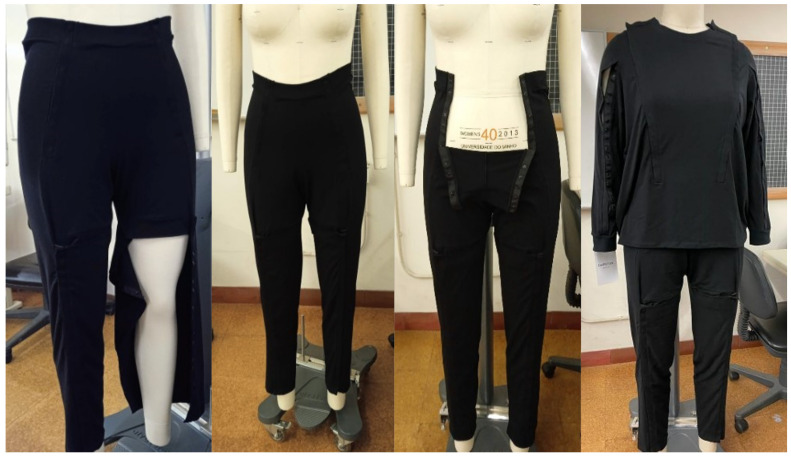
Prototype garment for the prevention of pressure injuries in people with reduced mobility and/or who are bedridden. Coimbra, Portugal, 2023.

**Table 1 ijerph-20-06773-t001:** Sociodemographic profile of specialist care nurses. Coimbra, Portugal, 2023.

Variables Evaluated	N	%
Gender		
Male	9	47.4
Female	10	52.6
Nursing Education		
Degree	5	26.3
Post-graduation/Specialization	5	26.3
Master’s Degree	7	36.8
Doctoral Degree	2	10.5
Clinical practice area		
Medical areas	15	78.9
Surgical areas	4	21.1
	M ± SD	CI 95%
Age (in years)	35.36 ± 9.68	30.70–40.03
Professional experience—overall (in years)	12.31 ± 8.96	7.99–16.63
Professional experience in the current ward (in years)	7.15 ± 8.61	3.00–11.31

Note: N: total sample; M: mean; SD: standard deviation; CI: confidence interval.

**Table 2 ijerph-20-06773-t002:** Usefulness assessment of the prototype garment modeling for PI prevention. Coimbra, Portugal, 2023.

Quantitative Assessment			Qualitative Assessment
Variables Evaluated	MD	CI 95%	Statements
Utility	4.0	3.03–4.55	*“The push buttons do not facilitate the exchange that I think it could be, maybe the goal that is to facilitate, no, but it makes sense these push buttons, it confuses at the time of dressing. […]” (Participant 01).* *“The locking system could be push ped on the simpler push, manipulated and fewer push. The other choice is to create a form of fasteners. The push buttons are too hard; this is not comfortable and will not make it easier to use […]” (Participant 12).* *“I don’t find the pants cuff adequate; without the cuff, it would make it easier to close, stitch closure through the opening. The push buttons are too many, it’s not practical. These push buttons do not help […]” (Participant 16).*
It is useful for my work	4.0	3.20–4.75	
It facilitates the performance of my task	4.0	3.15–4.70	
It allows me to be more effective/complete the task effectively	3.0	3.05–4.40	
It allows me to be more efficient/complete the task efficiently	3.0	3.10–4.50	
It meets the expected to respond to my task	4.0	3.25–4.55	
It allows me to complete the task	3.0	3.45–3.75	
It allows me to complete the task easily	4.0	2.85–4.29	
It allows me to complete the task quickly	3.0	2.55–4.05	
It allows me greater control over the task to be performed	4.0	3.55–4.75	
It helps me be more productive in my work	3.0	3.00–3.39	
It allows me to ensure more safety for the patient	4.0	3.20–4.90	
It allows me to respond to my needs	3.0	2.85–4.30	

Note: MD: median; CI: confidence interval.

**Table 3 ijerph-20-06773-t003:** Assessment of the “ease of use” and the prototype of the garment modeling for PU prevention. Coimbra, Portugal, 2023.

Quantitative Assessment			Qualitative Assessment
Variables Evaluated	MD	CI 95%	Statements
Ease of use	3.0	3.02–4.34	*“The push buttons are too many; it’s not practical. These pushes don’t help […]” (Participant 05).* *“The push buttons have already made it clear that they are too many and hard; it does not facilitate the use […]” (Participant 16).* *“Push buttons laterally to be able to open and have access to the front and back of the wearer would be interesting, the lateral opening should be whole. But there are too many push buttons […]” (Participant 10).*
It is easy to use	3.0	2.60–3.95	
It is simple to use	3.0	2.50–3.90	
It is user-friendly	3.0	2.70–4.20	
It requires few manipulations to achieve what is intended	3.0	2.25–3.80	
It allows flexible use according to my needs	4.0	2.80–4.39	
It does not require much physical effort in its use	4.0	2.80–4.39	
It does not require much mental effort in its use	4.0	3.55–4.79	
It allows me to conduct tasks in a logical and consistent sequence	4.0	3.55–4.79	
It is not associated with great possibilities of error in its use	3.0	3.15–4.50	
It allows me to correct any errors quickly and easily	5.0	4.05–5.20	

Note: MD: median; CI: confidence interval.

**Table 4 ijerph-20-06773-t004:** Assessment of the “ease of learning” and the prototype of the garment modeling for PU prevention. Coimbra, Portugal, 2023.

Quantitative Assessment			Qualitative Assessment
Variables Evaluated	MD	CI 95%	Statements
Ease of learning	5.0	4.35–5.57	*“That opening in the knee doesn’t make sense—for example, a knee wound. Regardless of the situation that occurred at the time, being bad and we must open everything, but the opening goes only to the knee […]” (Participant 02).* *“The push buttons, where we must put in and take out with force. It needs a manipulation that demands time […]” (Participant 10).* *“The push buttons, I have already made it clear that they are many and hard, it does not facilitate when dressing, it is confusing […]” (Participant 02).*
You quickly learn to use it	5.0	4.20–5.40	
You can easily learn to use it	5.0	4.10–5.35	
I quickly remembered how to use it	6.0	4.80–5.85	
I quickly became skilled at using it	5.0	4.10–5.55	
It does not take much prior knowledge to use it	5.0	4.65–5.75	
No written instructions are needed to use it	5.0	4.05–5.40	

Note: MD: median; CI: confidence interval.

**Table 5 ijerph-20-06773-t005:** Evaluation of the “satisfaction/intention to wear” question and the prototype of the garment modeling for PI prevention. Coimbra, Portugal, 2023.

Quantitative Assessment			Qualitative Assessment
Variables Evaluated	MD	CI 95%	Statements
I would be satisfied	4.0	3.20–4.55	*“When you lateralize the patient, you have an area that increases the pressure, and that opening is not good, or when it is in dorsal, I run the risk of hurting, I am thinking about the lateral push buttons […]” (Participant 4).* *The push buttons could have more space and be more malleable because I think it’s important not to be hard, not so hard. The push buttons would interfere with the safety of the user himself […]” (Participant 13).* *“Then, if you level up specifically the issue of spacing the push buttons. The push buttons could have more spacing and be more pliable because I think it’s important not to be hard, not to be so hard. The push buttons would interfere with the user’s own safety […]” (Participant 19).*
I would recommend it to colleagues	4.0	3.10–4.45	
It would allow me to perform the tasks I want	4.0	2.95–4.34	
It would be interesting for the performance of my tasks	4.0	3.20–4.65	
I would need to have it in my clinical practice	3.0	2.70–4.35	
It would be pleasant to use	4.0	3.05–4.90	
I would be comfortable with its use	4.0	3.50–4.70	
I would feel confident with its use	4.0	3.40–4.60	
It would give me security in its use	4.0	3.30–4.65	
The dimensions of the device are adequate	5.0	3.90–5.35	
The weight of the device is adequate	6.0	4.80–6.10	
Appearance is adjusted	5.0	4.35–5.75	
I would like to use it often	4.0	3.55–4.80	
It would be easy to adjust it for the performance of my tasks	4.0	3.45–4.85	

Note: MD: median; CI: confidence interval.

## Data Availability

The data presented in this study are available upon request from the corresponding author. Data will not be made publicly available due to the patenting process of the device under development. Acknowledgments: The authors would like to thank Impetus Portugal-Têxteis S.A., the Health Sciences Research Unit: Nursing (UICISA: E) of the Nursing School of Coimbra (ESEnfC), the Centre for Textile Science and Technology (2C2T) of the University of Minho, the International Iberian Laboratory of Nanotechnology (INL), and the Institute of Innovation on FibreBased Materials and Composites of the University of Minho for their involvement in the development of the sensor prototypes sent for evaluation by the ESEnfC nursing team.

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
