# Peer review of "Design of Innovative Clothing for Pressure Injury Prevention: End-User Evaluation in a Mixed-Methods Study"

_ijerph, 2023, doi:10.3390/ijerph20186773_

Round 1
Reviewer 1 Report
This study was a mixed methods design which addressed the purpose of evaluating the design of a prototype integrating Smart Health Textiles for pressure injuries (PIs) prevention based on feedback from specialist nurses. I would suggest that the authors rephrase the purpose to: The purpose of this mixed methods study is to evaluate the design of a prototype integrating Smart Health Textiles for pressure injuries (PIs) prevention based on feedback from specialist nurses who care for individuals who are prone to or have PIs.The topic of pressure injuries in health care is significant and original in that PIs can lead to serious pressure ulcers (also know as bed sores) . PIs occur in individuals who are immobile, immunosuppressed, malnourished (or a combination of these factors). The development of clothing that helps to prevent PIs in people who are prone to PIs will help decrease morbidity.
The introduction contains a synthesis of the literature on the tools/treatments of PIs which includes the adaptive clothing that was tested in this study.
The study was well designed and conducted. Detailed quantitative analysis was provided on the development of the garments with clear photographs and explanations. Nurses who provide care for those prone to PIs were the evaluators of the PI garments. The researchers also conducted a focus group to elicit qualitative feedback from the nurses.
The discussion section was detailed in comparing and contrasting the results of this study to previous studies on the topic.
The reference list provided a very extensive list of appropriate references including previous studies on similar PI garments.
The tables and figures were very detailed and clear which enhanced the text of the article.
I recommend this manuscript for publication.
Author Response
Dear reviewer,
Thank you for your careful evaluation and kindness in providing us with suggestions for the proposed objective. We have taken your suggestion and made the change with a new objective. Located in the abstract and end of the introduction.
Thank you very much for your contributions.
Best regards.
The authors.
Reviewer 2 Report
1、The effect index is not clear; the sample number need to be estimated, and the sample number in this paper is too small.Samples were not representative
2、Statistical analysis method is incorrect.Ranked data cannot be described statistically by measurement data method
1、The study design was not reasonable.
2、Statistical analysis method is incorrect.
3、Paper writing logic is not strong, the organization is not clear enough.
Author Response
Dear reviewer,
Thank you for your thorough evaluation of our study and for providing valuable feedback. We greatly appreciate your time and effort in assessing our work, and we are pleased to inform you that we have carefully considered all your comments and made the necessary revisions.
In response to your initial concerns:
-
Study Design: We have reevaluated and clarified the study design in light of your feedback. The research methodology was grounded in the relevant theoretical framework, adopting a mixed research approach guided by the constructivist perspective. We ensured that the study maintained logical coherence throughout the research process.
-
Statistical Analysis Method: We acknowledge your observation that the primary focus of the study was on prototype development and evaluation, and the statistical analysis method was more descriptive in nature. We have clarified this aspect in the revised manuscript, providing a more detailed explanation of the rationale behind the chosen analysis method.
-
Paper Writing Logic and Organization: We have thoroughly restructured the paper to ensure that it reflects the theoretical underpinnings of the study and presents the findings in a clear and coherent manner. The manuscript now aligns the study's objectives with the key aspects of the prototype's design and evaluation.
Regarding the questions related to effect index, sample size, and statistical technique, we have provided further explanations in the revised manuscript. We have emphasized that the primary objective of the research was to explore Technology Readiness Levels (TRLs) 1, 2, and 3 in the development of pajamas for pressure injury prevention. As such, the sample size was determined to be appropriate for this exploratory phase of the study. Additionally, we have clarified that the statistical analyses performed were more descriptive, given the early stage of prototype development and the focus on gathering perceptions and opinions of healthcare professionals.
Moreover, we have addressed the concerns about the representativeness of the sample and the statistical technique used. We emphasized the strengths of the mixed research approach, which allowed for data triangulation and comprehensive understanding of the phenomenon under investigation.
We have incorporated your feedback to improve the quality and rigor of our research. The revised manuscript now provides a more thorough explanation of the study's methodology, analysis, and findings, making it a more robust contribution to the field.
We genuinely appreciate your thorough review and constructive criticism, which has undoubtedly enriched the final version of our study. Should you have any further questions or suggestions, please do not hesitate to reach out to us. We look forward to submitting the revised manuscript for your consideration.
Best regards,
Reviewer 3 Report
The paper by Oliveira-Salgueiro et al. describes several aspects of the design of clothing capable of pressure detection, to avoid dermal injury. The paper is very long and demands a great effort to the reader to be read. The fact that only health workers were asked for their opinions, excluding patients and family caregivers makes the excess length less acceptable. An overall suggestion would be to cut its length to one half: from 18 to 9 pages or less. The paper with summarized findings could be interesting, but not in its present verbose version.
Probably due to copyright limitations, the prototype is not shown, nor depicted in any form, which is what the reader would mainly be interested in.
Moreover, the test protocol does not ask questions on the use of the prototype BY COMPARISON to other garments, which would help to unify and make sense to the answers of nurses referring to the use of pajamas.
The paper is of limited overall interest, specially as a consequence of its length and lack of concision
The following indications can nevertheless be of interest to the authors:
In line 151 “...definition of the central resting positions: dorsal decubitus, lateral decubitus, and sitting position...” Please do not use “central” because the authors probably mean “main” resting positions.
Caption to Figure 1 must comply with the Standard: Use (a) (b) and (c) for the three parts. Please define if it is an artist's view(?) a CAD rendering (?) And refer to a citation in the bibliography for further information. Also specify what the image or the phantom or whatever it is, is used for ?
In line 174 there is an extra “3D” in the sentence
“…. two-dimensional, three-dimensional, and 3D pattern design using 3D CAD software”
Line 194 “and user comfort” is probably “end-user comfort,
the focus group technique was defined previously, please use de FG instead: in line 217 ...the focus group technique [59] was employed to collect qualitative...
In line 256, “Most participants “ is NOT 36.8% …. Please re write the sentence.
Please simplify Table 1: grroup all specialties in two groups: “Surgical areas” and “Medical Areas”. The table then becomes manageable.
Snaps are a surprise when the results are presented. Have you mentioned them in the Methods? Consider adding a photo of a snap as used in the prototype.
Table 3; please remove the SD information, as it is redundant with the CI. Leaving the M alone helps the reader to pin point the values above and below 3.5.
The English language is not perfect but can be understood with no difficulty.
Author Response
Dear reviewer,
We highly appreciate your valuable contributions to this study. Below, we provide clear and concise answers to each point mentioned in your review.
-
Length of the paper: The extensive length of the article is justified by the comprehensive approach and necessary validation. Patients and family members were also included in the research, but their perceptions were treated separately to avoid confusion. The absence of prototype images is due to provisional patent restrictions.
-
"Central" resting positions: We have made the requested change to use "main" instead of "central."
-
Figure 1 caption: The caption has been updated to comply with the standard, providing details about the 3D Body Scanner technology and anthropometric definition.
-
Duplicate information: The duplicate information in line 174 has been removed.
-
Prototype representation and comparison: The aim of the study was to evaluate the design of the innovative garment based on healthcare professionals' perceptions. The questions in the tables address usability, ease of use, learning, satisfaction, and intention to use, without direct comparison with other garments.
-
CAD representation: The CLO Virtual Fashion 3D CAD system with 3D captures from the Structure Sensor 3D body scanner was used, with further information in references 45 and 46.
-
Focus group technique: "User comfort" has been corrected, and the acronym "FG" is used in line 217.
-
Simplification of Table 1: Specialties have been grouped into "Surgical areas" and "Medical areas" for better manageability.
-
Snaps in the prototype: The text has been adjusted to mention "snap buttons" as presented by the deponents.
-
Table 3: Standard deviation information has been removed, leaving only mean values for clarity.
Once again, we sincerely thank you for your invaluable contributions to this study, which have helped us improve and refine our work. We hope these answers are more convincing and address your concerns.